# OpenReview forum: "A Generative Approach to LLM Harmfulness Detection with Red Flag Tokens"
_ICLR.cc/2025/Workshop/BuildingTrust — BuildingTrust_

### Official Review · Reviewer_WSee · 2025-02-28
**This work introduces adding special "red flag" token into the model and training the model to prepend generation of unsafe content with said token. They highlight that this method is more robust to black-box and grey-box adversarial attacks.**

**Rating:** 7
**Confidence:** 4

**Review:**

This work is written clearly, and the method introduced in this paper seems original.

Why is the index where <rf> is placed is sampled? Would it be beneficial to decide spans of harmful responses and only their input <rf> token?
Figure 2 subplots should be larger to increase their readability and ease of understanding the results.
Pros:
* Multiple models were evaluated, and multiple benchmarks were utilized to ensure that the model's performance didn't go down.
* I think such tokens could be used with other safety-increasing methods, making LLM more robust.
* Authors evaluated their method on multiple types of attacks.
Cons:
* The part of the dataset with harmful content was very small.
* It would be nice to see how this method influences the model performance after full fine-tuning instead of LoRA. Due to LoRA being known for forgetting less information when fine-tuning [1], it could be the case here that when this method was applied with full finetuning, the changes in the MMLU and other datasets could be larger.

[1] Biderman, Dan, et al. "Lora learns less and forgets less." Transactions on Machine Learning Research (2024).

---

### Official Review · Reviewer_8G66 · 2025-03-01
**Comments**

**Rating:** 8
**Confidence:** 4

**Review:**

Strengths:

1. The paper introduces an innovative method for assessing response safety by introducing a red flag token. This approach avoids rejections that could harm the quality and distribution of responses to harmless queries, thereby preserving the model's utility.
2. It simplifies the evaluation of the model's robustness and, when combined with a classifier, reduces the likelihood of correlated failures.
3. Comprehensive experiments and analyses are provided to support the paper’s conclusions.

Weaknesses:
1. While the approach significantly outperforms the baseline on grey-box assumptions, its performance on white-box is worse than the baseline.
2. The impact of adding the red flag token on utility might be sensitive to the fine-tuning data used.

---

### Official Review · Reviewer_yFpn · 2025-03-02
**Review of A Generative Approach to LLM Harmfulness Detection with rf Tokens**

**Rating:** 5
**Confidence:** 3

**Review:**

## Summary
The paper introduces a red flag token (⟨rf⟩) for large language models (LLMs) to detect harmful outputs by training the model to generate ⟨rf⟩ whenever harmful content is produced or about to be produced, enabling it to act as a generative classifier of harmfulness. Through extensive experiments, the authors demonstrate that this approach maintains model utility, improves robustness against adversarial attacks (e.g., pre-filling and sampling), and can be stored in a LoRA module for post-hoc application to fine-tuned models.

## Strengths

1. Theoretical and Empirical Validation:
Provides a well-designed loss function and extensive experiments across multiple models and attack scenarios

2. Robustness to Adversarial Attacks:
Demonstrates strong defense against realistic attacks like pre-filling and sampling, as well as generalization to long contexts.

## Weaknesses

1. Hyperparameter Sensitivity:
The approach requires careful tuning of hyperparameters, which may limit its practicality without extensive resources.
2. Dependence on Training Data:
The effectiveness of ⟨rf⟩ relies on the quality and diversity of the training data, which may not cover all harmful scenarios.
3. Lack of comparison with classifier method:
The paper does not quantitatively compare the ⟨rf⟩ method to a standalone harmfulness classifier in terms of performance metrics and trade offs.

---

### Decision · Program_Chairs · 2025-03-04

**Decision:**

Accept

**Comment:**

The paper introduces an innovative red flag token (⟨rf⟩) to detect harmful outputs in LLMs, demonstrating robustness against adversarial attacks while maintaining model utility. However, its effectiveness is sensitive to hyperparameters and training data quality, lacks direct comparison with classifier-based methods, and its impact under full fine-tuning remains unexplored.